# Progressive Data Dropout: An Adaptive Training Strategy for Large-Scale Supervised Learning

## Abstract

Common training strategies for deep neural networks are computationally expensive, continuing to redundantly train and evaluate on classes already well-understood by the model. A common strategy to diminish this cost is to reduce data used in training, however this often comes at the expense of the model's accuracy or an additional computational cost in training. We propose *progressive data dropout* (PDD), an adaptive training strategy which performs class-level data dropout from the training set as the network develops an understanding for each class. Our experiments on large-scale image classification demonstrate PDD reduces the total number of datapoints needed to train the network by a factor of 10, reducing the overall training time without significantly impacting accuracy or modifying the model architecture. We additionally demonstrate improvements via experiments and ablations on computer vision benchmarks, including MNIST, Fashion-MNIST, SVHN, CIFAR, and ImageNet datasets.

## 1 Introduction

Deep neural networks have made a significant impact on a broad range of applications over the last decade. However, these networks are notoriously data-intensive, often requiring significant computational power and large datasets in order to properly train for optimal performance. This can become problematic for many real-world applications as the computational expense of training these networks will often prevent them from being adopted.

Many optimization techniques have arisen to address this problem in training - from reducing neurons to subsampling data. In this work, we focus on reducing the computational time and cost needed to fully train any deep network, without modifying the model and while utilizing the entire training dataset.

In order to properly set the discussion, we need to define some terms that are used throughout this paper. First, we define a *datapoint* as a single data sample that is sent through the network during the training process. For example, if you were to train a network using a dataset of 10 samples for 5 epochs, you would have used 50 datapoints to train that network. So our datapoint calculations are a collection of the number of samples that were sent to the network during training rather than the number of unique data samples it was provided. This is because networks often need to see examples of a class multiple times before understanding them, Secondly, since our proposed method modifies the training process, the term epochs no longer applicable as we will not iterate over the entire dataset after dropping data. Instead, we will use the term *training rounds* to indicate how many times we iterated over the remaining training set when training the network.

Now, we consider one simple research question: *Can data be dropped during training once it is well-understood by a deep leaning model?* We propose *Progressive Data Dropout* (PDD), an adaptive training process which leverages the network's understanding of the data to determine when data should be dropped from the training process. In comparison with existing data dropout techniques which focus on identifying "important" samples, we instead evaluate the simple case of full and partial class-removal.

The main contributions of our simple adaptive training strategy, PDD, include:

- Reduces time and computational resources required for training.
- Model-agnostic implementation which works for any supervised single-label classification task.
- Data-agnostic implementation which does not preprocess or examine the data for sample quality, balance, etc.
- Provides an inherent stopping criterion for training models on sufficiently large datasets.

Further sections motivate the use of PDD for large-scale image classification tasks and describe the design of this simple strategy. We propose PDD to be complimentary of existing regularization techniques, as well as recent learning/training strategies such as continual learning, curriculum learning, and others.

## 2 RELATED WORK

There are a breadth of optimization techniques for the training deep neural networks, motivated by the traditionally large data requirements of deep models. Given that we are proposing a generalized strategy for optimizing training, this section discusses several types of solutions - learning techniques, dropout for network regularization, and dropout of data during training.

### 2.1 CURRICULUM & CONTINUAL LEARNING

Several learning approaches are fundamentally similar in objective to our proposed method, though they take a drastically different approach. Curriculum learning, for example, aims to slowly introduce difficult aspects of the problem by training the network from the easiest samples to the hardest samples. Lyu & Tsang (2019) introduce a curriculum loss for learning with noisy labels. By introducing a loss function that is bounded and able to select samples during network training, the authors are able to combine continual and robust learning.

Meta-learning similarly aims to quickly adapt to new tasks by improving the learning algorithm itself, improving computational bottlenecks as well as generalization. Finn et al. (2017) However, both meta-learning and curriculum learning suffer from occasional catastrophic forgetting of the original learned task. Continual learning aims to prevent the catastrophic forgetting of the original learned task Veniat et al. (2021). More recently, Co2L (Cha et al., 2021) contrastive continual learning proposed a contrastive learning objective which learned and preserved representation through distillation. These curriculum and continual learning strategies are complimentary to PDD in that our proposed method can be leveraged toward their respective objectives.

### 2.2 DROPOUT FOR NETWORK OPTIMIZATION & REGULARIZATION

While our proposed approach is focused on dropping *data*, the term *dropout* more often applies to the dropping of neurons during the training process for regularization of a neural network (Srivastava et al., 2014). Network dropout regularization techniques K C et al. (2021) and pruning approaches (Tanaka et al., 2020) reduce the size of network during training in response to input stimuli.

Adaptive dropout for training deep neural networks (Ba & Frey, 2013) overlaid a binary belief network on top of a neural network allowing the network to adapatively regularize the network by selectively setting parts to zero. Dropconnect (Wan et al., 2013) regularized networks by dropping randomly selected weights instead of randomly selected activations. Curriculum dropout (Morerio et al., 2017) showed that the a fixed neuron dropout probability was sub-optimal and instead implemented a time scheduler for updating the dropout probability. An energy-based dropout proposed by EDropout (Salehinejad & Valaee, 2021) used an energy based loss to find the best pruning to apply to the original neural network. In application to different network architectures, Dropout-GAN dropped connections between a generator and multiple discriminators in a GAN in order to ensure diversity of generated samples, avoiding mode collapse (Mordido et al., 2018).

While dropout is effective for regularization, these methods often require significant modification of the network itself in order to be adopted. In comparison, our proposed approached is entirely model-agnostic, modifying only the number of samples used in training instead of the network or

data itself. We therefore do not compare with these types of approaches in our experiments, as they are complementary to PDD rather than comparative.

## 2.3 DATA DROPOUT

Most similar to our proposed method are approaches which drop data samples during the process of training a neural network. Some such approaches focus on data augmentation, in an effort to improve the quality of the data fed to the network and model accuracy. Generalized Dropout (Rahmani & Atia, 2018) for example is a method of data augmentation, dropping random pixels from an image in order to generate additional samples for training.

Other methods for data dropout emphasize importance sampling, aiming to first identify quality samples and then drop the lower quality samples for remaining training epochs (Katharopoulos & Fleuret, 2018). Data Dropout (Wang et al., 2018) removes samples from training after they're deemed unfavorable in the first epoch, saving time in further training rounds. Similarly, DropSample (Yang et al., 2016) applied this concept for Chinese character classification. Subsequently a Greedy DropSample (Yang et al., 2020) generalizes the method even further by using a greedy algorithm, temporarily dropping data as needed for training acceleration, but retaining it for use if the model requires. In a similar fashion, dataset summarization techniques which strategically subsample the dataset have been demonstrated to outperform full-set models (Wang et al., 2020).

Recently Han et al. (2020) dropped highly negative predictions iteratively through the utilization of influence functions. This helped remove highly noisy or out of date labels from the training set. Similarly, Dynamic Training Data Dropout (DTDD) (Zhong et al., 2022) drops samples deemed noisy after several epochs, with specific application to noise-robust deep face recognition.

In contrast to these methods, our *progressive data dropout* takes cues from the model while training to determine when to drop a majority of a class. While each of these existing approaches is similar to our proposed PDD, we do not compare with those which are so domain-/application-specific as to not be applicable to standard image classification benchmarks, including DTDD, DropSample for character classification, and those removing noisy labels. Such methods are complimentary to PDD rather than comparative as they can be used in combination for their application-specific tasks.

## 3 PROGRESSIVE DATA DROPOUT

In this work, we present *Progressive Data Dropout* (PDD), a novel training strategy which combines data dropout with a residue component to train networks. Like most current data dropout techniques, PDD can be paired easily with most networks since it requires no network modifications. However, unlike most other data dropout techniques, PDD progressively removes data from the training set as the network develops an understanding for classes. By removing data from the training set, it allows us to create subsets of the data for a network to train on, speeding up the training process. To do this, we utilize dropout score in conjunction with a residue component to control which subset of the dataset is provided to the network.

Figure 1, and further demonstrated through our extensive experiments, displays a general view of the data used in training a neural network. Baseline training strategies have a constant number of datapoints since all data is used in every training round, whereas comparative data dropout approaches drop datapoints during the training process. This figure demonstrates the significant reduction in data feasible in training a network. The following subsections detail PDD components and training strategy in more detail.

## 3.1 DROPOUT SCORE

An important component to PDD's strategy is selecting a dropout score since it is responsible for determining when data should be dropped from the training examples. When determining PDD's dropout score, there are two important factors to consider. First, the dropout score should be a metric that accurately reflects the performance of the individual classes in the network. For example, in our experiments we used f1-score when evaluating classification models since it depicts how well the network is performing on a per-class basis. Secondly, once a metric has been selected as the dropout score, we then need to select an appropriate threshold. Determining the threshold should be treated

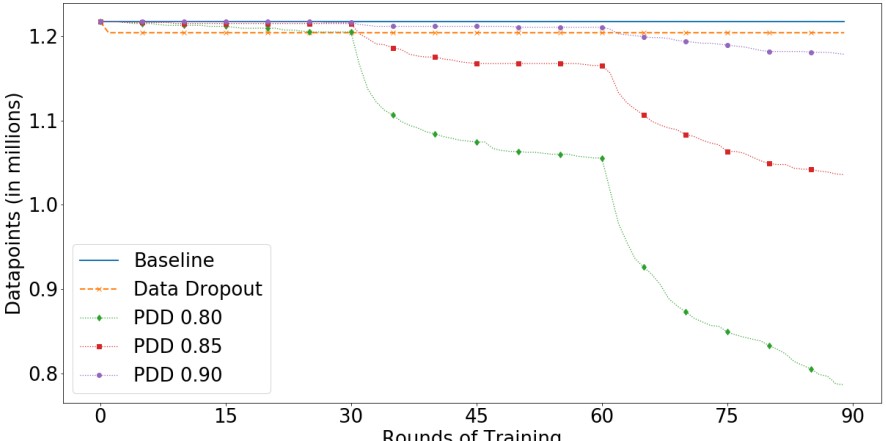

Figure 1: Visualization of the amount of training datapoints used training ResNet34(He et al., 2016) on ImageNet (Deng et al., 2009). Best viewed in color.

as an extremely sensitive hyper-parameter which can have significant effects on the network training process. If the threshold value is set too low, the network will often not have a firm grasp of the data being dropped, resulting in the eventual forgetting of the data. If the threshold value is set too high, many of the training data points may never be dropped, severely reducing the benefit provided by PDD. Unfortunately, like many other hyper-parameters, there is no universally applicable threshold value as it is highly dependant on your both the metric and the dataset. To help determine an appropriate metric and its threshold, we recommend experimenting with a smaller data subset first.

## 3.2 RESIDUE

Intuitively, and reinforced in our experiments, fully removing a class from the training dataset causes the network to suffer catastrophic forgetting due to the loss only penalizing the network for predicting the removed class. Rather than modifying the loss, which would be considered a network modification, we instead introduce a residue component into PDD to prevent the catastrophic forgetting problem. This residue component controls what percentage of randomly-selected training examples associated with a class are left behind when that class crosses the dropout score threshold. Unlike the dropout score threshold, we empirically show through our experiments that leaving behind ten percent of a class is usually enough to encourage the network to remember it. It is important to note that this residue percentage can be increased; however this will come at the cost of training time.

## 3.3 RESIDUE SWAPOUT

When leaving a small percentage of data behind as residue, alleviating the catastrophic forgetting problem, it presents a new problem where networks will often easily over-fit on the residue examples. To address this problem, we also introduce a residue swapout component to determine how often residue training examples are swapped out for newer ones. While this does add additional computational costs, it also helps prevent the network from over-fitting on the residue examples, especially over long periods of training.

## 3.4 WARM-UP PERIOD

We also introduce a warm-up period component into PDD. This parameter specifies how long the network is allowed to train before our training strategy is allowed to begin. This parameter is to ensure that the network's feature space has been well established before we begin removing data. Typically, this parameter is mostly used for lower-dimensionalty data where a network can converge on a dropout score too quickly.

### 3.5 TRAINING STRATEGY

Once the components of PDD have been established, we begin PDD by creating a mask of the entire datasets' labels before beginning network training. This mask is used throughout the training process to determine which subset of the dataset to provide to the network during a single training round. Once the mask has been created, PDD begins the network training process with the entire mask on, providing the network with the entire dataset. At the end of a training round, a dropout score is calculated for each individual class in the dataset. If a class has a dropout score that is above the provided threshold, and assuming the warm-up period has also been met, we assume the network has developed an understanding for that class and its presence can be safely reduced in the training data. In order to reduce the class' presence, we update the mask to exclude all but randomly selected residue examples of the class, creating a new data subset to use in the next round of training. PDD concludes the network training process when either every class has exceed the provided dropout score threshold or when the maximum rounds of training has been reached. Overall, PDD reduces the number of datapoints that are needed to train the network, which in turn reduces the overall training time of a network. In addition, PDD can provide a clear point to when your network should be stopped with training and help determine classes the network has difficulty converging on.

## 4 EXPERIMENTS

In order to demonstrate the effectiveness of PDD, we first conduct image classification experiments in this section and then explore ablation studies for PDD's components in the subsequent section. For the purposes of all of our experiments, we selected f1-score as our dropout score metric. Also, unless otherwise specified, we used the following values for PDD's components: a residue of .10, a residue swapout of 1 training round, and a warmup period of 5 training rounds.

### 4.1 IMAGE CLASSIFICATION BENCHMARKS

We first conducted experiments on lower dimensionality datasets with well-established classification models to show the efficacy of PDD. We trained Resnet-18 on MNIST (LeCun et al., 2010), FashionMNIST (Xiao et al., 2017), SVHN (Netzer et al., 2011), and CIFAR10 (Krizhevsky, 2009) using a single RTX 8000 GPU. During training, we used the following hyper-parameters: batch sizes of 32, a standard categorical cross-entropy loss, a standard SGD optimizer with learning rate of 0.01, and a max number of training rounds of 20. Finally, we report the accuracy of the model in relation to the dataset's test set. Table 1 shows the results of our experiments in comparison with other data dropout techniques.

### 4.2 DROPOUT SCORE THRESHOLD EXPERIMENT

After conducting our benchmark experiments, we then demonstrate the effect the dropout score threshold has on PDD. Table 2 shows the results of this experiment. Besides MNIST, our experiments show that a higher droput score yields a better network performance at the cost of training time. Our experiments also demonstrate a faster training time due to PDD concluding the network training process early. This is because every class has exceed the provided dropout score threshold thus ending the training of the network.

### 4.3 LARGE-SCALE IMAGE CLASSIFICATION

We extend our experiments to a large-scale image classification problem. Our experiments show the result of our training method when applied to the ImageNet (Deng et al., 2009) dataset on a well-established classification model. To do this, we trained ResNet-34 on the ImageNet dataset using 8 V100 GPUs, applying a different training strategy to each run. During training, we followed the standard PyTorch training procedure (PyTorch, 2022) which included: batch sizes of 32, a standard categorical cross-entropy loss and a SGD optimizer with a momentum of 0.9, weight decay of 1e-4, and a learning rate of 0.01. As the network training progressed, we reduced the learning rate by a factor of 10 for every 30 rounds of training that occurred. The results of this experiment are shown on Table 3 which evaluated the performance of the network training methods on the ImageNet

| Method | Dataset | Model | Acc. | Time (mins) | RT | # Datapoints |
|---|---|---|---|---|---|---|
| Baseline | MNIST | ResNet18 | 0.99 | 0:13:45 | 20 | 1,080,000 |
| GD | MNIST | custom | 0.99 | - | 250 | - |
| DropSample | MNIST | custom | 0.99 | 0:25:00 | 800 | - |
| **PDD** | MNIST | ResNet18 | 0.99 | 0:04:12 | 6 | 324,000 |
| Baseline | FashionMNIST | ResNet18 | 0.91 | 0:14:03 | 20 | 1,080,000 |
| **PDD** | FashionMNIST | ResNet18 | 0.88 | 0:04:25 | 7 | 334,265 |
| Baseline | SVHN | ResNet18 | 0.92 | 0:17:08 | 20 | 1,318,640 |
| DataDropout | SVHN | ResNet152 | 0.99 | *0:28:20* | 50 | 2,107,811 |
| **PDD** | SVHN | ResNet18 | 0.91 | 0:05:04 | 6 | 395,592 |
| Baseline | CIFAR-10 | ResNet18 | 0.77 | 0:11:59 | 20 | 900,000 |
| IS | CIFAR-10 | ResNet28 | *0.95* | 5:30:00 | 50,000 | - |
| DropSample | CIFAR-10 | custom | *0.92* | 0:18:20 | 24 | - |
| DataDropout | CIFAR-10 | ResNet110 | 0.95 | *4:10:00* | 500 | *21,901,200* |
| **PDD** | CIFAR-10 | ResNet18 | 0.71 | 0:04:27 | 9 | 324,011 |

Table 1: Comparison of metrics of various data dropout techniques. Compares a baseline training strategy, GeneralizedDropout (GD) (Rahmani & Atia, 2018), ImportanceSampling (IS) Katharopoulos & Fleuret (2018), Greedy DropSample (Yang et al., 2020), DataDropout Wang et al. (2018), and and PDD (our proposed Progressive Data Dropout method, with 0.85 F1 dropout score). Estimated values are *italicized*, and clarified in the appendix. Missing values are noted as '-', and all other values are as reported in the original works.

| Method | Dataset | Dropout Score | Accuracy | Time (mins) | RT | # Datapoints |
|---|---|---|---|---|---|---|
| Baseline | MNIST | - | 0.99 | 0:13:45 | 20 | 1,080,000 |
| PDD | MNIST | 0.85 | 0.99 | 0:04:12 | 6 | 324,000 |
| PDD | MNIST | 0.95 | 0.99 | 0:04:20 | 6 | 324,000 |
| Baseline | FashionMNIST | - | 0.91 | 0:14:03 | 20 | 1,080,000 |
| PDD | FashionMNIST | 0.85 | 0.88 | 0:04:25 | 7 | 334,265 |
| PDD | FashionMNIST | 0.95 | 0.90 | 0:08:24 | 17 | 597,705 |
| Baseline | SVHN | - | 0.92 | 0:17:08 | 20 | 1,318,640 |
| PDD | SVHN | 0.85 | 0.91 | 0:05:04 | 6 | 395,592 |
| PDD | SVHN | 0.95 | 0.91 | 0:06:21 | 9 | 477,939 |
| Baseline | CIFAR-10 | - | 0.77 | 0:11:59 | 20 | 900,000 |
| PDD | CIFAR-10 | 0.85 | 0.71 | 0:04:27 | 9 | 324,011 |
| PDD | CIFAR-10 | 0.95 | 0.75 | 0:07:25 | 14 | 544,913 |

Table 2: F1-score PDD experiment on various datasets using ResNet18 (He et al., 2016). RT stands for rounds of training. PDD settings shared across runs: Warmup period of 5 rounds, residue of .10, and a swapout period of 1 round. Dashed values are not applicable.

validation dataset. We additionally provide a visual which shows the number of datapoints over the network's training in Figure 1.

| Method | F1 Dropout Score | Validation Accuracy | Time (hours) | RT | # Datapoints |
|---|---|---|---|---|---|
| Baseline | - | 0.73 | 8:30:12 | 90 | 109,544,850 |
| Data dropout (Wang et al., 2018) | - | 0.79* | - | 60* | 108,375,212* |
| PDD | 0.80 | 0.70 | 7:31:38 | 90 | 94,957,921 |
| PDD | 0.85 | 0.72 | 8:08:12 | 90 | 103,996,872 |
| PDD | 0.90 | 0.73 | 8:26:22 | 90 | 108,570,798 |

Table 3: Comparison of run times on ImageNet (Deng et al., 2009). RT stands for rounds of training * Denotes the value was taken or calculated from the original paper.

## 5 ABLATION STUDIES

To assess the effectiveness of each component of our proposed method, the following studies demonstrate the effect of removing our residue, swapout, and warm-up features from PDD. We conduct these studies on the lower-dimensionality data using the same training specifications that were mentioned Section 4.1. In addition, for these studies, PDD components were set to the following values: dropout score of 0.85, a residue of .10, residue swapout of 1 training round and a warmup of 5 training rounds unless that component was turned off, in which case its value was set to 0.

### 5.1 RESIDUE COMPONENT

In this study, we looked at the effects the residue component had on PDD network training. Table 4 shows the results of these studies. As shown in the table, the networks trained on FashionMNIST and CIFAR-10 performed significantly worse if there is no residue component. For the MNIST and SVHN networks, PDD ended the training process immediately after the warm-up period. This meant that all of the classes exceed the dropout score after the warmup period ended, resulting in the residue and no residue experiments being treated exactly the same.

| Method | Dataset | Accuracy | Time (mins) | RT | # Datapoints |
|--------|---------|----------|-------------|-----|--------------|
| Residue | MNIST | 0.99 | 0:04:12 | 6 | 324,000 |
| No Residue | MNIST | 0.99 | 0:04:15 | 6 | 324,000 |
| Residue | FashionMNIST | 0.88 | 0:04:25 | 7 | 334,265 |
| No Residue | FashionMNIST | 0.10 | 0:04:23 | 7 | 329,410 |
| Residue | SVHN | 0.91 | 0:05:04 | 6 | 395,592 |
| No Residue | SVHN | 0.92 | 0:05:05 | 6 | 395,592 |
| Residue | CIFAR-10 | 0.71 | 0:04:27 | 9 | 324,011 |
| No Residue | CIFAR-10 | 0.15 | 0:04:25 | 9 | 319,497 |

Table 4: Residue PDD experiment on various datasets using ResNet18. RT stands for rounds of training. PDD settings: Dropout score of 0.85, residue of 0.10 where applicable, swapout period of 1 training round, and a warmup period of 5 rounds.

### 5.2 SWAPOUT COMPONENT

In this study, we looked at the effects the residue swapout component had on PDD network training. Table 5 shows the results of these studies. Similar to the last study, networks trained on FashionMNIST and CIFAR-10 suffered from performance drops when removing the swapout component, although significantly less than when compared to the residue component. Again, PDD ended the training process of the MNIST and SVHN networks immediately after the warm-up period, resulting in the swapout experiments being treated the exact same.

| Method | Dataset | Accuracy | Time (mins) | RT | # Datapoints |
|--------|---------|----------|-------------|-----|--------------|
| Swapout | MNIST | 0.99 | 0:04:12 | 6 | 324,000 |
| No Swapout | MNIST | 0.98 | 0:04:14 | 6 | 324,000 |
| Swapout | FashionMNIST | 0.88 | 0:04:25 | 7 | 334,265 |
| No Swapout | FashionMNIST | 0.86 | 0:04:29 | 7 | 334,265 |
| Swapout | SVHN | 0.91 | 0:05:04 | 6 | 395,592 |
| No Swapout | SVHN | 0.93 | 0:05:03 | 6 | 395,592 |
| Swapout | CIFAR-10 | 0.71 | 0:04:27 | 9 | 324,011 |
| No Swapout | CIFAR-10 | 0.70 | 0:04:26 | 9 | 319,950 |

Table 5: Swapout PDD experiment on various datasets using ResNet18. RT stands for rounds of training. PDD settings: Dropout score of 0.85, residue of 0.10, swapout period of 1 training round where applicable, and a warmup period of 5 rounds.

## 5.3 WARMUP COMPONENT

Finally, we studied the effects that the warmup component had on PDD network training. Table 6 shows the results of these studies. Unlike the previous studies, all networks but ones trained on CIFAR-10 suffered from performance drops when removing the warmup component. However, it is important to note that having no warmup component did improve training time significantly. As for the networks trained on CIFAR-10, since it is unlikely to drop a class within the first 5 rounds of training due to the complexity of the data, the warmup period component did not matter.

| Method | Dataset | Accuracy | Time (mins) | RT | # Datapoints |
|---|---|---|---|---|---|
| Warmup | MNIST | 0.99 | 0:04:12 | 6 | 324,000 |
| No Warmup | MNIST | 0.97 | 0:00:42 | 1 | 54,000 |
| Warmup | FashionMNIST | 0.88 | 0:04:25 | 7 | 334,265 |
| No Warmup | FashionMNIST | 0.87 | 0:02:01 | 5 | 138,886 |
| Warmup | SVHN | 0.91 | 0:05:04 | 6 | 395,592 |
| No Warmup | SVHN | 0.87 | 0:01:58 | 3 | 146,314 |
| Warmup | CIFAR-10 | 0.71 | 0:04:27 | 9 | 324,011 |
| No Warmup | CIFAR-10 | 0.72 | 0:04:18 | 9 | 311,847 |

Table 6: Warmup PDD experiment on various datasets using ResNet18. RT stands for rounds of training. PDD settings: Dropout score of 0.85, residue of 0.10, swapout period of 1 training round, and a warmup period of 5 rounds where applicable.

## 6 CONCLUSIONS

In this work, we propose *Progressive Data Dropout* (PDD), a new training optimization strategy for deep learning networks. Through extensive experimentation on well-established classification networks and benchmarks, we demonstrate an effective training strategy which reduces data over network training on both large-scale and small-scale image classification tasks. By reducing the data over network training, we are able to reduce the overall time needed to train a network while also showing which classes a network is having difficulty learning.

### 6.1 LIMITATIONS

Although we demonstrate the effectiveness of PDD, there are limitations that need to be addressed. One major limitation of our proposed training method is that it is designed with classification networks in mind. This is mainly due to our dropout score working inherently with a classification problem rather than other tasks such as regression. Another major limitation of our training strategy is that we assume that the training dataset is relatively large and balanced, which means our strategy would likely cause performance issues for problems such as few-shot or anomaly detection. Finally, our current implementation creates a copy of dataset labels to use as the mask, which could be expensive for problems with a high label cost such as segmentation and multi-label tasks.

### 6.2 FUTURE WORK

This proposed PDD implementation randomly selects the residue examples that are left behind when a class crosses the dropout score threshold. In future works, this randomized residue component could be replaced with a more deterministic component, such as a method which finds low confidence samples, in order to improve the overall classification performance of the network. However, when adding the deterministic component, it is important to ensure that the overhead of the new method maintains a lower computational cost than just training with the entire dataset.

In conjunction with the deterministic residue component, an interesting avenue of exploration is a more dynamic residue component. With a dynamic residue component, the network could have different levels of residue for each class depending on their performance. However, there are some roadblocks to implementing this, such as determining a proper metric tracking and computational overhead.

Though PDD focuses on data-level techniques, it can easily be paired with other data dropout or network modification techniques in order to further improve the overall performance of a network, including those identified in section 2. Since PDD tracks how well a network is performing on a per-class basis, that information could be leveraged with other techniques to assist with class understanding and retention.

Finally, our proposed training method could be extended to other tasks in computer vision, language processing, and general classification. While PDD could be extended to other single label problems with relatively minimal changes, multi-label problems such as semantic segmentation or even multi-label classification would require some extension in order to identify when a sample could be appropriately dropped. In a large enough dataset, a policy could be established for justifying the removal of data samples containing only well-understood classes, for example.

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

## A APPENDIX

### A.1 CALCULATION OF ESTIMATED VALUES

In our paper we compare our proposed PDD approach with related data-dropping methods on 5 image classification benchmark datasets. However, not all of these cited works provided results

on all datasets, nor code to replicate their results. Instead, we were able to estimate some of their missing information via their described algorithm. In an effort to be as transparent as possible, we include these calculations here, organized by table/experiment.

### A.1.1 TABLE 1 DATADROPOUT ESTIMATIONS

The DataDropout method proposed by Wang et al. (2018) reported 1220 "unfavorable" training samples in CIFAR-10 found after the first epoch. Therefore the calculated number of datapoints for the 500 training rounds noted in the paper for ResNet-110 was (1 epoch of the full CIFAR-10 training set) + (full CIFAR-10 training set - 1220) * (500-1 training rounds) = (45,000) + (45,000-1,200) * 499 = 21,901,200. If it took our baseline ResNet-18 model 0.6 minutes per training round on CIFAR-10, we can conservatively estimate training this ResNet-110 model on fewer datapoints to be 0.5 * (number of training rounds = 0.5 * 500 = 250 minutes = 4 hours and 10 minutes on our single GPU.

Similarly, for SVHN, DataDropout reported 24,261 "unfavorable" training samples found after the first epoch. Therefore the calculated number of datapoints for the 50 training rounds in the paper for ResNet-152 was (1 epoch of the full SVHN training set) + (full SVHN training set - 24,261) * (50-1 training rounds) = conservatively 65,932 + (65,932-24,261)*49 = 2,107,811. Their reported accuracy was 98.53. If it took our baseline ResNet-18 model 51 seconds per training round on SVHN, we can conservatively estimate training rounds = (51 - 51/3) * 50 rounds of training = 34 * 50 = 1,700 seconds = 28 minutes and 20 seconds.

### A.1.2 TABLE 1 IMPORTANCE SAMPLING ESTIMATIONS

The Importance Sampling method proposed by Katharopoulos & Fleuret (2018) reported sampling the CIFAR-10 dataset in a variable fashion, where each iteration resamples the dataset. For this reason, and because it was not reported, the number of datapoints is not calculable. Their graph in their figure 3 approximates the error close to 0.95. This graph reports run time of conservatively 20,000 seconds, which is   5 hours and 30 minutes.

### A.1.3 TABLE 1 DROPSAMPLE ESTIMATIONS

The Greedy DropSample method proposed by Yang et al. (2020) reported stopping the training of a neural network when the "accuracy stays at one for 10 epochs." This work used custom CNNs, rather than out-of-the-box models for the experiments we compare with. For both the MNIST and CIFAR-10 experiments, graphs are reported to show values for total number of dropped samples for each round of training a model. Due to the number of training rounds, we are unable to fully estimate a number of datapoints for this related work.

For MNIST, their CNN model is shown in their figure 6 as a 16-layer convoluational network, consisting of: [convx2 + pool + dropout] x 2 + [conv + pool] + [dropout + dense] x 3. They trained on MNIST for approximately 800 epochs, according to their figure 7 graph, taking approximately 1,500 seconds, which is 25 minutes. They report an error rate at 0.4%, and therefore report 0.995 accuracy for their model.

For CIFAR-10, the model used was the DAWN Benchmark winning entry, which consisted of a 3-layer CNN with residual connections. They ran this model for 24 epochs, taking conservatively 1,100 seconds based upon their graph in figure 11 - this is 18 minutes and 20 seconds. Approximated from the same figure, we estimate error rate at 8%, and therefore report estimated 0.92 accuracy for their model.

### A.1.4 TABLE 1 GENERALIZEDDROPOUT ESTIMATIONS

The GeneralizedDropout method was proposed by Rahmani & Atia (2018). They ran a custom CNN model consisting of 5-five layers: 3 convolutional and 2 fully-connected. This model is trained on 15,000 images randomly sampled from MNIST. It is trained for 50 epochs without data dropout, then for 200 epochs "using different basis matrices for the random data dropout." Due to the random data dropping in their proposed method, without a reported run time or count of data samples dropped per round, we are unable to estimate these values.

