# OpenReview forum: "Progressive Data Dropout: An Adaptive Training Strategy for Large-Scale Supervised Learning"
_ICLR.cc/2023/Conference — Submitted to ICLR 2023_

### Official Review · Reviewer_tLVt · 2022-10-13

**Confidence:** 3
**Correctness:** 3
**Technical Novelty And Significance:** 2
**Empirical Novelty And Significance:** 2
**Recommendation:** 3

**Clarity, Quality, Novelty And Reproducibility:**

In terms of presentation (clarity + quality) I enjoyed how Section 5 ablates on each element of the method. In terms of novelty I did not understand "we do not compare with those which are so domain-/application-specific as to not be applicable to standard image classification benchmarks" and would appreciate more detail there.

**Strength And Weaknesses:**

Strengths: Method is well motivated and an ablation is performed on each of their design decisions.

Weaknesses:
- I found the comparisons to the baselines to be very unclear. For instance, in Table 1 there are various different architectures, values of epochs, and # of datapoints, such that it is not really clear how to compare properly to baselines. So, while the method is sound, it would be extremely useful to see a comparison where the task, architecture, RT, and datapoints are matched to make a comparison.

Comments & Questions:
- May be good to discuss related work such as [1,2].
- How would the method be extended to large datasets which are often unlabelled?

[1] https://arxiv.org/abs/2107.07075
[2] https://arxiv.org/abs/2206.14486

**Summary Of The Paper:**

This paper proposes a technique for removing data points for standard supervised learning problems. Their best performing method partially drops data from a class with a sufficient F1 score, and shows that they can train on ~6M total less data points while only reducing ImageNet accuracy by 1 percentage point.

**Summary Of The Review:**

The paper is well motivated and performs careful ablations, however, the comparison to prior methods was extremely hard to follow and discussion of key related work was missing.

---

> ### Author Response · Authors · 2022-11-09
> **Thanks & response**
>
> Thank you for your feedback!
> We will revise the results for clarity, controlling for time as well as accuracy. Rounds of training are intentionally varied in our experiments - our approach will not require as many rounds of training or data points as existing approaches. We could revise Section 4 to clarify this further.
>
> We will incorporate your noted references for related work - thank you for these! This approach removes 20% of ImageNet data while maintaining accuracy, which provides a good comparison with our approach - we aim to remove far more data during training. This will update Sections 1 and 2.
>
> Your question regarding unlabeled data is excellent, however it may be beyond the scope of this paper - significant adaptations of our approach would be needed for unsupervised learning, so we have focused on supervised learning.
>
> We could revise Section 2.3 to further clarify the rationale for not comparing with some of the related domain-specific approaches. For example, from our references, DropSample is designed specifically for Chinese character recognition, where the model they use is looking for specific strokes for certain characters. In this case, they are unable to run their methods on image classification benchmarks which do not contain Chinese characters, or similarly utilize baseline architectures without significant modifications. Would this clarification add to the paper?

---

> > ### Comment · Reviewer_tLVt · 2022-11-15
> > **Thanks.**
> >
> > Thanks for the response. After also reading the replies from other reviewers I will leave my score as is.

---

> > > ### Author Response · Authors · 2022-11-16
> > > **Follow up?**
> > >
> > > Thanks for your response. Our questions are meant to strengthen the study, are you able to respond to these?

---

### Official Review · Reviewer_De4X · 2022-10-24

**Confidence:** 5
**Correctness:** 3
**Technical Novelty And Significance:** 2
**Empirical Novelty And Significance:** 1
**Recommendation:** 3

**Clarity, Quality, Novelty And Reproducibility:**

The method was simple and presented clearly. The idea to drop almost entire classes seems novel. The code is in the supplement.

**Strength And Weaknesses:**

Strengths:
-- The data dropout idea is well-motivated - time or "datapoints" spent on different classes does not have to be equal. Some classes may add little value once they are learned and it makes sense to discard them and save time.

-- The experiments are performed on multiple datasets of different sizes and compared with various previous methods.

Weaknesses:
-- The results on MNIST, CIFAR and SVHN do not convincingly demonstrate the effectiveness. The time spent is drastically reduced but the accuracy is also hurt, sometimes by large margins. The comparison are done usually without controlling the time or accuracy so it is hard to draw useful conclusions.

-- The baseline ResNet-18 accuracy of CIFAR-10 is at 77%. This should be around 88% instead for ResNet-18 with a reasonable recipe. It is thus hard to judge the reliability of the results given the baseline may be too suboptimal.

-- On the larger scale ImageNet experiments, the saved time is too minimal (less than 1%) when the accuracy is maintained.

**Summary Of The Paper:**

The paper proposes a progressive data dropout (PDD) framework that gradually discards the majority of the samples for already-accurate classes, in order to speedup the training. There are other design choices including warmup, residue and swapout. Experiments on MNIST, CIFAR, SVHN and ImageNet are performed.

**Summary Of The Review:**

Given the insignificance of the results I recommend rejection.

---

> ### Author Response · Authors · 2022-11-09
> **Thanks &. response**
>
> Thank you for your feedback!
> We will revise the results for clarity, controlling for time as well as accuracy.
>
> Concerns regarding the baseline ResNet-18 accuracy on CIFAR: the noted accuracy in this review is correct for CIFAR-10 on either smaller versions of ResNet (e.g. networks with 0 or 2 convolutional layers), or the larger ResNet models (e.g. ResNet18 or ResNet110) with modifications such as node dropout, data augmentation, or significant batch sizes. We have reported on ResNet-18 without these network modifications and without adding data during training in order to set our baseline. Per additional reviewer feedback, we aim to expand Table 1 to include further architectures, however we would not augment the data during training as some of these related approaches (e.g. adding data, when our goal is to reduce training data). Would you recommend we clarify this in the paper?
>
> “…ImageNet …saved time is less than 1%”:
> For these experiments, we leveraged 8 V100 GPUs, so while a 1% decrease in resources seems small, this would scale with the hardware available for larger problems. While the time saved is minimal in the current environment, there are situations where you would have less available hardware, for example maybe you only have 2 V100s instead of 8, and that 1% would scale into a significantly larger time save. Also in comparison with the noted related approaches, PDD additionally identifies the more easily-learned and more difficult classes during training.
> The 1% saved time noted is for an accuracy which is exactly equal to the baseline. We could clarify in the discussion of Table 3 the benefit of PDD to applications with limited hardware where a slight sacrifice in accuracy may be preferable due to the savings in time and computational resources. For ImageNet, a 1% decrease in validation accuracy saved ~5% of time and used nearly 5 million fewer datapoints (noted in Table 3).

---

### Official Review · Reviewer_9Ung · 2022-10-24

**Confidence:** 3
**Correctness:** 1
**Technical Novelty And Significance:** 2
**Empirical Novelty And Significance:** 2
**Recommendation:** 3

**Clarity, Quality, Novelty And Reproducibility:**

**Clarity**

Paper was readable and the method was understandable. However, the experimental setup was quite confusing (covered in weaknesses).

**Novelty**

The method seems novel.

**Reproducibility**

Reproducibility unclear. The author's seem to have not reproduced their baseline properly, however they did provide code which seems readable. Have not assessed in detail.


**Strength And Weaknesses:**


##### Strengths

- Method is intuitive and easy to understand.
- Problem relevance is apparent, reducing training time would have immediate impact.

##### Weaknesses

- Table 3 shows that the author's reimplementation of their baseline was flawed. The results of the existing data dropout paper are reproducible.
- It's very unclear what Tables 1 & 2 are supposed to convey. There are no comparable comparisons in any of these experiments. The authors should fix an architecture and see how many data points it takes to train to the _same accuracy_. As it is, with variations in architecture, final model performance, and # of data points, we can't draw any conclusions from these tables.

**Summary Of The Paper:**

This paper works on the problem of progressive data pruning, in which you attempt to reduce the number of images (in absolute quantity) during training. They do this by subsampling according to class using a dropout score, which is related to the performance of the network on a particular class in the training set. The introduce a hyperparameter (training threshold), which determines when data should be dropped wrt the dropout score.

**Summary Of The Review:**

I vote to reject.

Their primary experiments are flawed. As is, we cannot to properly assess the efficacy of their method (see Weaknesses for suggestions to improve).

---

> ### Author Response · Authors · 2022-11-09
> **Thanks & response**
>
> Thank you for this helpful feedback!
> We would like to provide clarity in the results by fixing to a single model or set of models in Table 1 specifically. We would report results beyond ResNet18 to include ResNet28, ResNet110, ResNet152, as well as several custom CNNs from related works. Do these seem like worthwhile additions to Table 1, and would you recommend any further clarifications?
>
> We anticipate these clarifications will address concerns with training to the same accuracy - we will show our approach on the same model(s) and more easily demonstrate accuracy consistent with the related works. In this case, Table 1 will be able to emphasize the time, rounds of training, and datapoints for each approach. We may retain Table 2 to demonstrate the effect of the tunable parameter Dropout Score, as there are applications where a slight sacrifice in accuracy would be useful if time & computational resources are saved.
>
> “Table 3 shows that the author’s reimplementation of their baseline was flawed” - can you please expand on this?  The results of Data Dropout are reproducible, and are noted in the table. In regards to the baseline model, Our baseline accuracy matches those provided by a Pytorch Resnet-34 model with Imagenet weights shown in the link https://pytorch.org/vision/stable/models.html.
>
> One further clarification: are there any other concerns regarding reproducibility/clarity or correctness?

---

> > ### Comment · Reviewer_9Ung · 2022-11-21
> > **Concerns not properly addressed**
> >
> > After reviewing the updated Table 1, it seems my concerns have not addressed. The author needs to show that their method improves one of {performance, training time, amount of training data needed, etc.} while keeping all other quantities fixed, otherwise it's hard to draw concrete conclusions. The experimental methodology is flawed. I will keep my score as is.

---

### Official Review · Reviewer_YwiM · 2022-10-29

**Confidence:** 4
**Correctness:** 4
**Technical Novelty And Significance:** 3
**Empirical Novelty And Significance:** 2
**Recommendation:** 3

**Clarity, Quality, Novelty And Reproducibility:**

- The paper was very clear and easy to comprehend.
- The quality of the paper is not very high since the results don't clearly show the advantage of this approach.
- The approach is novel to my knowledge.
- I did not attempt to reproduce any of the experiments in the paper.

**Strength And Weaknesses:**

# Strengths
- The paper is very well written and easy to comprehend.
- The problem tackled by the paper is very interesting and important to the community as it could speed up training significantly.
- The approach presented is novel to my knowledge and explores an interesting direction that I haven't seen before.

# Weaknesses
- The results are difficult to parse. It's really hard to compare the different approaches since they are trained for a different number of rounds using potentially different models.
- There also seems to be a significant cost in accuracy to using PDD.
- Why do the alternative data-dropping strategies achieve higher test accuracy than even the baseline? It seems like the baseline should be the best approach since it has access to the most data.
- There should be more experiments with different model architectures.
- There are no standard errors in the results.

**Summary Of The Paper:**

This paper proposes a novel, adaptive strategy to drop classes from the dataset during training based on the f1 score of the model. They show that their approach, Progressive Data Dropout (PDD), is able to achieve similar test/val accuracies while using significantly lower data and time to train on image classification tasks.

**Summary Of The Review:**

While the approach considered in the paper is interesting and novel to my knowledge, the results leave much to be desired. The baseline approach considered seems considerably worse than expected, especially compared to the alternatives from the literature. There also seems to be a significant cost to using PDD in the final test/val accuracy of the model.

---

> ### Author Response · Authors · 2022-11-09
> **Thanks & response**
>
> Thank you for your detailed feedback!
> We would like to provide clarity in the results by fixing to a single model or set of models in Table 1 specifically. We would report results beyond ResNet18 to include ResNet28, ResNet110, ResNet152, as well as several custom CNNs from related works. Do these seem like worthwhile additions to Table 1, and would you recommend any further clarifications?
>
> “There are no standard errors in the results” - can you please expand on this? We could clarify Section 4 further with experiment details, if any are found missing.
>
> As far as training for different number of rounds - this is an intentional byproduct of our approach, we will not need to train for as many rounds as the related approaches because we are dropping data and watching the F1 score. We could make this clearer in the paper in Section 3.5 to specify rounds of training rather than general training time. Would this address your concern?

---

### Decision · Program_Chairs · 2023-01-20

**Decision:**

Reject

**Justification For Why Not Higher Score:**

Reviewers are unanimous in rejecting the paper as experiments are unconvincing with respect to benefits over the baseline.  The author response does not address these concerns.

**Justification For Why Not Lower Score:**

N/A

**Metareview: Summary, Strengths And Weaknesses:**

This paper proposes to use class-level data dropout to accelerate training of neural networks.  Reviewers unanimously move to reject the paper, with a serious concern being experiments that show a nontrivial cost in accuracy to using the proposed method.  The author response does not present any additional results, so these concerns are unresolved and reviewers maintain their rating.  The Area Chair agrees with the reviewer consensus.